# Long-Term Alleviation of the Functional Phenotype in Chlorophyll-Deficient Wheat and Impact on Productivity: A Semi-Field Phenotyping Experiment

**DOI:** 10.3390/plants12040822

**Published:** 2023-02-12

**Authors:** Andrea Colpo, Sara Demaria, Costanza Baldisserotto, Simonetta Pancaldi, Marian Brestič, Marek Živčak, Lorenzo Ferroni

**Affiliations:** 1Department of Environmental and Prevention Sciences, University of Ferrara, Corso Ercole I d’Este 32, 44121 Ferrara, Italy; 2Institute of Plant and Environmental Sciences, Slovak University of Agriculture, Trieda A. Hlinku 2, 949 76 Nitra, Slovakia

**Keywords:** acclimation, antenna size, chlorophyll fluorescence, electron transport, photoprotection, photosynthesis, photosystem II, wheat

## Abstract

Wheat mutants with a reduced chlorophyll synthesis are affected by a defective control of the photosynthetic electron flow, but tend to recover a wild-type phenotype. The sensitivity of some mutants to light fluctuations suggested that cultivation outdoors could significantly impact productivity. Six mutant lines of *Triticum durum* or *Triticum aestivum* with their respective wild-type cultivars were cultivated with a regular seasonal cycle (October–May) in a semi-field experiment. Leaf chlorophyll content and fluorescence parameters were analysed at the early (November) and late (May) developmental stages, and checked for correlation with morphometric and grain-production parameters. The alleviation of the phenotype severity concerned primarily the recovery of the photosynthetic-membrane functionality, but not the leaf chlorophyll content. Photosystem II (PSII) was less photoprotected in the mutants, but a moderate PSII photoinhibition could help control the electron flow into the chain. The accumulation of interchain electron carriers was a primary acclimative response towards the naturally fluctuating environment, maximally exploited by the mature durum-wheat mutants. The mutation itself and/or the energy-consuming compensatory mechanisms markedly influenced the plant morphogenesis, leading especially to reduced tillering, which in turn resulted in lower grain production per plant. Consistently with the interrelation between early photosynthetic phenotype and grain-yield per plant, chlorophyll-fluorescence indexes related to the level of photoprotective thermal dissipation (pNPQ), photosystem II antenna size (ABS/RC), and pool of electron carriers (Sm) are proposed as good candidates for the in-field phenotyping of chlorophyll-deficient wheat.

## 1. Introduction

Plant phenotyping is prospected as a key tool to increase crop productivity and understand the impact of biotic and abiotic stresses on plant fitness [1]. Paramount data can be obtained through phenotyping experiments in controlled environments, such as climate chambers or phenotyping units. A further necessary step is validation of results in real agricultural contexts [1,2,3]. Semi-field experiments are an intermediate step between controlled laboratory environments and the open field. This approach is quite popular in (agro)ecology to verify predictions emerging from laboratory findings in an easily monitored near-natural environment, where the experimental material is exposed to realistic climatic conditions [4]. Moreover, the semi-field experiment can allow the timely integration of in-field measurements with laboratory analyses without any significant delay due to sample transportation.

Photosynthesis is the process that enables the plant to produce the building blocks for its growth and development by converting solar energy into chemical bonds. Consequently, increasing a crop’s photosynthetic efficiency plays a key role in improving productivity. Photosynthetic phenotyping is allowed by various techniques also in the field, for example, analysing the leaf gas exchange or chlorophyll fluorescence emission [5]. The latter is especially advantageous because it is quick to perform and non-invasive, allowing the collection of a large amount of data while preserving samples [6].

Most of the work produced in recent decades in photosynthetic plant phenotyping has converged on the hypothesis that decreasing the leaf chlorophyll content could help improve crop productivity [7,8]. A lower chlorophyll content could theoretically reduce the amount of thermally dissipated light energy and increase the light availability for the lower leaves, potentially leading to higher photosynthetic efficiency and, consequently, biomass production [7]. Recent studies especially have reported that some mutants or transformants with an overall reduced leaf-chlorophyll content [9,10], or a specific reduction in chlorophyll *b* [11] or a depletion in specific antenna proteins [12], could potentially have a growth benefit. One may expect that many decades of breeding would have indirectly favoured plants with a lower leaf-chlorophyll content. However, no significant decrease in chlorophyll content emerged from a detailed comparison of 26 winter wheat cultivars released during 1940–2010 in China [13].

Chlorophyll-depleted mutants often exhibit peculiar features in their photosynthetic membranes, among which is an altered stoichiometry between photosystem I and II (PSI and PSII); this, in turn, causes an imbalance in the excitation rate between PSI and PSII, an abnormal organization of the thylakoids, changes in the spatial distribution of the two photosystems, and reduced control of photosynthetic electron flow, resulting in delayed/reduced growth and increased sensitivity to environmental factors [14,15,16,17,18,19]. The chlorophyll-deficient wheat lines used in this study are derived from wild-type cultivars of *Triticum durum* L. (durum wheat LD222) and *Triticum aestivum* L. (bread wheat Novosibirskaya 67, NS67), and have been extensively characterized for the structure, organization, and functionality of their photosynthetic apparatus: ANDW-7A, ANDW-8A and ANDW-7B (7A, 8A, 7B) for durum wheat, and ANBW-4A, ANBW-4B and ANK-32A (4A, 4B, 32A) for bread wheat (Table A1). At the seedling stage of indoor cultivation, these mutants are characterized by a reduction in total chlorophyll content of between 40% and 70%, along with an increase in chlorophyll *a*/*b* ratio; the most severe mutant phenotype was found in 7B and 32A [14,15]. However, the differences flatten out in the later stages of development in the mutants grown indoors [14,15]. In contrast, plants grown outdoors (March–June) tended to retain the typical characteristics of the chlorophyll-depleted phenotype, particularly in durum-wheat mutants [14]. It was proposed that their growth in the open field may exacerbate an already compromised control of linear electron flow (LEF), caused by an insufficient cyclic electron flow (CEF). This in turn causes a reduced capacity for energy dissipation, ATP synthesis, and photoprotection. The combination of these alterations is responsible for a tendency toward the over-reduction of the acceptor side of PSI, potentially leading to a significant rate of PSI inactivation [20]. A chronic overreduction of the electron transport chain can be extremely harmful for plants and even threaten their survival [21]. Nevertheless, the chlorophyll-depleted wheat mutants cope with the electron-flow deregulation through the accumulation of PSI end electron acceptors (ferredoxin and ferredoxin-NADP^+^-oxidoreductase, FNR) and, in addition, the enhancement of the electron-sink alternative to photosynthesis and photorespiration [15,16]. These compensative adjustments inevitably affect the biomass yield of the mutants, which is generally lower than that of their wild-type lines [15]. In an indoor experiment, the plant growth under a fluctuating-light regime aggravated this condition specifically in the 7B and 4B mutants, very probably because of recurrent photosynthetic electron bursts [22]. However, and surprisingly, the 4A mutant seemed to benefit instead from the fluctuating-light regime, and its biomass yield was even higher than that of NS67.

Up until now, the relevant information about this collection of mutants for potential field cultivation can be summarized as follows:(1)The phenotype severity tends to decrease during the life cycle of the plants cultivated indoors; i.e., the leaf chlorophyll content recovers to wild-type-like levels (phenotype alleviation) [14];(2)The phenotype alleviation is incomplete outdoors with a short (March–June) life cycle [14];(3)The phenotype alleviation can be incomplete indoors if plants are cultivated under fluctuating light [15,16].

Nevertheless, the contrasting results obtained with mutants sharing similar degrees of chlorophyll depletion (e.g., 7B and 7A) make the outcome of an open-field cultivation basically unpredictable.

The present work aims to elucidate how outdoor cultivation of chlorophyll-depleted wheat mutants affects the photosynthetic characteristics at the early (November) and late (May) developmental stages when wheat is cultivated in a regular seasonal cycle. A semi-field approach was chosen to allow timely integration of in-field fast chlorophyll-fluorescence measurements with in-laboratory modulated chlorophyll-fluorescence analyses. Based on observations from previous work, two alternatives were considered possible: (a) the direct outdoor sowing in October could exacerbate the phenotype of chlorophyll-depleted mutants because of the everchanging environmental conditions, including light fluctuations; (b) alternatively, the overwintering cultivation, necessary for the tillering phase, allows for an effective long-term acclimation, for instance emphasizing the compensatory mechanisms previously described. In both cases, it was interesting to assess whether the mutants were affected by a loss in grain yield and whether the functional phenotypic parameters already proposed for the characterization of the mutants [15,16] could be validated and/or implemented with other chlorophyll-fluorescence-derived parameters.

## 2. Results

### 2.1. Pigment Content in Young and Mature Plants

Total chlorophyll content and chlorophyll *a*/*b* molar ratio are markers for the phenotype severity in both durum and bread-wheat mutants [14,15,16]. The two wild-type lines, LD222 and NS67, were the richest in chlorophyll, followed by two intermediate mutants (7A and 8A for durum, 4A and 4B for bread wheat) and the more severe phenotype of 7B for durum wheat and 32A for bread wheat (Figure 1). All genotypes showed an increased chlorophyll content in the leaves of the mature plants compared to the young, although much more marked in bread wheat (almost double) than in durum wheat (+15%, Figure 1). The increase was homogeneous across genotypes, so the severity gradient was maintained.

The chlorophyll *a*/*b* ratio is inversely proportional to the size of the PSII antenna system, the light-harvesting complex II (LHCII), which hosts most chlorophyll *b*. In young durum-wheat mutants, it was significantly higher than in the wild-type LD222, while bread wheat 4A and 4B were intermediate, between NS67 and 32A (Figure 1). Plant maturation tended to maintain the differences between the mutant and the wild-type lines in both species; only in the bread-wheat mutant 32A, was the chlorophyll *a*/*b* ratio approx. 30% higher in the mature, compared to the young, plants.

The relative variations in the total chlorophyll/carotenoid molar ratio were roughly similar to the trends observed for chlorophyll (*a* + *b*), but the range of variation was obviously much narrower (Figure 1).

### 2.2. Irradiance-Dependence of PSII Quantum Yields

Light-response curves of PSII quantum yields of photochemistry [Y(II)], constitutive energy loss [Y(NO)] and thermal dissipation [Y(NPQ)] were recorded in samples collected in the field and immediately transported to the laboratory.

Starting from the reference value in the dark-acclimated state (corresponding to *F_V_/F_M_*), Y(II) underwent a progressive decay at increasing irradiance, which depends on the strength of electron sinks, primarily the Calvin–Benson–Bassham cycle. The curves obtained from mutants and wild-type lines were nearly overlapping (Figure 2A–D).

Y(NO) can highlight defects in the regulation of the plastoquinone redox state [15,23]. The Y(NO)-irradiance dependence in the wild-type lines showed a typical behaviour, with a peak at the lowest light intensity related to the functioning of only LEF (Figure 2E–H); subsequently, at higher irradiance, it decreased because of the activation of the CEF [24,25,26,27] and the related regulatory processes known as photosynthetic control operating at the cytochrome *b*_6_*f* complex [28,29]. The major divergences in Y(NO) were observed in young plants of mutant durum wheat, particularly 7B, reflecting the inefficiency in regulating LEF (Figure 2E,G). At the mature stage, all durum-wheat mutants were able to regulate the electron poise properly (Figure 2F). In bread wheat 4A and 4B, LEF was generally well regulated; the leaves of young 32A tended instead to diverge from the wild type at the highest irradiances, and this physiological trait worsened in the mature plants (Figure 2H). Persisting high Y(NO) values up to intermediate irradiances marked a delay in the activation of CEF. After the peak, Y(NO) decreased but maintained higher values than NS67, indicating that in the mature plants of 32A, the control of the electron flow could not be recovered to a wild-type level (Figure 2H).

Y(NPQ) reflected the phenotype severity in young durum-wheat mutants and showed their lower efficiency in inducing thermal dissipation compared to LD222, whereas, in the mature plants, the tendency to approach a wild-type behaviour was evident (Figure 2I,J). Among bread-wheat lines, only 32A exhibited a defective induction of thermal dissipation, which was not improved in the mature plants (Figure 2K,L).

### 2.3. Quenching Analysis and PSII Photoprotection

Stern–Vollmer NPQ is the standard parameter to quantify the overall thermal dissipation in quenching analysis [30,31,32]. Because NPQ equals the Y(NPQ)/Y(NO) ratio [33,34], the differences among wheat lines were emphasised (Figure 3A–D). In young plants, the initially tiny differences between wheat lines were progressively magnified, up to the maximum NPQ value (NPQ_MAX_), exactly matching the phenotype-severity series (Figure 3A,C). In mature durum wheat, the differences between the genotypes became negligible (Figure 3B); in mature bread wheat, the ineffective NPQ induction was confirmed in 32A, although it was less severe than in young plants (Figure 3D). The mature mutant 4B was, surprisingly, the most effective in inducing NPQ in response to the increasing light intensity (Figure 3D).

Lower values of NPQ are associated with a lower photoprotective ability against PSII photodamage [35,36,37]. In chlorophyll-deficient wheat mutants, a conclusion could not be drawn with a sufficient degree of confidence for mutants developing NPQ to the same extent as the wild-type lines. To discriminate between the photoprotective and non-photoprotective components of NPQ, the parameter pNPQ calculated according to the “qP_d_ model” was introduced by Ruban and Murchie [38,39]. The photoprotection offered by NPQ in young plants was maximum in the two wild-type lines and in 4A, independent of the plant age (Figure 3E–H). Less pNPQ was developed by all young durum-wheat mutants, as well as by 4B and 32A (Figure 3E,G). In the leaves of mature plants, the alleviation of the impaired pNPQ phenotype was complete in 7A and 8A, but not even partial in the other mutants (Figure 3F,H). Despite mature 7B and 32A tending to approach NPQ_MAX_ values identical or closer to their wild-type lines, their capacity for PSII photoprotection remained very low (Figure 3F,H). Although the mature 4B was capable of inducing higher NPQ than NS67, it did not gain a superior PSII photoprotective capacity (Figure 3H).

To understand the impact of a lower degree of PSII photoprotection on overall PSII activity, we analysed the photochemical quenching. The parameter 1-qP provides information on the fraction of closed PSII—or the excitation-energy pressure inside active PSII centres [40]. Very interestingly, all 1-qP curves were nearly overlapping, even in the young plants (Figure 3I–L).

### 2.4. Fast-Chlorophyll-a-Fluorescence Transient and JIP-Test Parameters

Analysis of the fast chlorophyll *a* fluorescence transients (OJIP) was performed directly in the field. In young plants, the OJIP transients were clearly influenced by the plant genotype (Figure 4A,C). *F_0_* values in the two wild-type lines were obviously higher, with the biggest differences found for the durum-wheat mutants (Figure 4A). For a qualitative comparison of the mutants, the fluorescence transients were first double normalized between *F_0_* (step O) and *F_M_* (plateau P, Figure 4E–H)). The two inflections at approx. 2 ms (step J) and 30 ms (step I) were variably affected in the mutants. The geometrical features of the transients were used to calculate technical parameters (Sm), quantum yields and probabilities (φPo, ψETo, ψREo and δREo) and specific energy fluxes per reaction centre (ABS/RC, TRo/RC, ETo/RC, REo/RC, DIo/RC), according to the model by Strasser et al. [41] and Stibert and Govindjee [42]. The definitions of the parameters are reported in Appendix B (Table A2), and the results are comparatively shown in the diagrams of Figure 5.

In the young plants, the initial exponential rise of chlorophyll *a* fluorescence was slightly slower in the mutants compared to their wild-type counterparts (Figure 5A,C); this feature can be attributed to a smaller PSII-antenna size in the mutants, which was quantified by significantly lower ABS/RC values (Figure 5A,C). Such slower increase was uniform among mutants, without apparent relation to phenotype severity as assessed by pigment analysis. The same samples also showed a tendency to higher φPo (Figure 5A,C). In young durum-wheat mutants and in 32A, Sm, ψETo, ψREo and δREo were enhanced, as compared to the corresponding wild-type lines (Figure 5A). In particular, higher ψREo and δREo indicated a more abundant relative pool of electron carriers downstream of PSI [16,43]. In the semi-field condition of this experiment, the same mutants also underwent an increase in ψETo—related to the probability that an electron is accepted by plastoquinone, and thus the pool size of the latter [41,42,44] (Figure 5A). This response was in line with the tendency to accumulate more electron carriers per PSII-PSI chain unit, as evidenced by generalized higher Sm values (Figure 5A).

In the young plants of mutants, all energy fluxes, except REo/RC, per active PSII centre were depressed, in comparison with the corresponding wild-type lines (Figure 5A,C). Given the dependency of these parameters on the normalized initial slope of the OJIP transient, such a response clearly descended from the smaller PSII-antenna size of the mutants (Figure 4A,C and Figure 5A,C). The most affected flux was that related to the untrapped-energy dissipation per active PSII (DIo/RC), which was reduced between 25 and 50%, depending on the mutant. The decrease in DIo/RC was easily related to the seeming gain in PSII photochemical efficiency, φPo. A noticeable parameter was REo/RC: in young durum-wheat mutants and 32A it was similar to the wild-type values, while in 4A and 4B it was lower (Figure 5B). REo/RC describes the energy flux from Q_B_ to PSI end acceptors; high values were consistent with the increased probability of electron transfer from Q_B_ to PSI end acceptors (ψREo) in these mutants (Figure 5B).

Measurements performed at the end of the season revealed a generalized alleviation of the mutant functional phenotype (Figure 5B,D). In particular, *F_0_* and the specific energy fluxes became more similar between wild-type and mutant lines, compared to the young plants. However, some peculiarities were found with respect to the electron-transfer probabilities and Sm. In durum-wheat mutants the number of interchain electron carriers increased considerably, following the gradient of phenotype severity (Figure 5B). This response was absent in the intermediate mutants of bread wheat (4A, 4B), whereas it was apparent in 32A (Figure 5D). ψREo and δREo similarly increased in the same mutants.

### 2.5. Productivity Estimation, Morphometric Measurements and Correlation Analysis

At the end of the 2020 and 2021 seasons (third week of May) morphometric and productivity measurements were performed after harvesting the grains. Plant height, ear number and length did not show major differences among the genotypes, except for a reduced plant height in 8A, 4A and 32A in comparison to their wild-type lines; moreover, the ear in 32A was significantly shorter compared to the other bread-wheat genotypes (Table A3). Conversely, the tillering degree (number of stems per plant) was very sensitive to the genotype, being reduced in all mutants (Figure 6C,G). The number of grains per ear was reduced only in durum-wheat mutants 7A and 7B (Figure 6A,E). The weight of a single grain was more affected by the severity of the phenotype, being lower in the most severe mutants, 7B and 32A (Figure 6B,F). The product of the number of grains per ear from the weight of a single grain and the tillering degree resulted in the synthetic parameter termed “single-plant productivity”, which summarizes the potential for plant productivity for each wheat line (Figure 6D,H). Compared to LD222, in 8A the productivity was one-third less, in 7A it was halved, and in 7B more than halved (60% decrease, Figure 6D). Likewise, the bread-wheat mutants were generally less productive than NS67, with productivity reduced by ca. 10% in 4B, 25% in 4A and 60% in 32A (Figure 6H).

The ambition of crop photosynthetic phenotyping is to support predictions of plant productivity. A correlation matrix was built to check the presence of significant correlations between photosynthetic parameters recorded in young plants and productivity/morphometric parameters obtained at the end of the season, when mature grains were harvested (Figure 7). Morphometric parameters such as plant height, spikelet number and ear length did not display any significant correlation with the photosynthetic parameters. Conversely, several correlations were found between photosynthetic and grain-productivity parameters. The individual morphometric/productivity parameters showed weak correlations with the pigment parameters, with only two exceptions (total chlorophylls vs. tillering degree; chlorophyll/carotenoid ratio vs. single-grain weight). Much more interesting was the strong correlation between all pigment parameters and the synthetic index of the productivity of a single plant. A considerable number of correlations were found with fluorometric parameters as well. The number of grains per ear was negatively correlated with Y(NO)_MAX_ (i.e., the highest value of Y(NO) recorded during a light-response curve); the single-grain weight was negatively correlated with Sm, and the tillering degree with pNPQ. However, taking again the single-plant productivity as the most informative of the plant-performance indexes, the most sensitive fluorometric parameters were NPQ_MAX_, pNPQ, Sm, ABS/RC, and TRo/RC.

## 3. Discussion

In previous experiments, the indoor cultivation of the chlorophyll-deficient wheat mutants evidenced their tendency to recover a wild-type phenotype with respect to the leaf chlorophyll content, emphasizing the impact of the genomic context, particularly regarding which subgenome (A or B) hosted the mutated locus and the condition of hexaploidy (bread wheat) vs. tetraploidy (durum wheat) [14,15,16]. Moreover, the different sensitivity of the mutants to light fluctuations revealed the relevance of the environment in shaping their phenotype [15,16]. The metabolic-energy investment in compensatory responses capable of limiting the detrimental effects of a disturbed photosynthetic electron flow was proposed to be the main cause for a lower biomass accumulation. Whether this conclusion could also hold true under a regular life cycle in the field needed testing, looking at phenotypic traits in association with the grain yields: under a naturally fluctuating environment, does the phenotype recover to wild type? Is the mutant-severity series confirmed? Does the mutation have a significant impact on grain production?

In general, the young plants confirmed previous findings, showing the phenotype severity in the order 7B > 7A = 8A > LD222 and 32A > 4A = 4B > NS67 [14], with the functional defects matching the chlorophyll-depletion level quite well. A tendency of the mutants to adjust their leaf carotenoid content occurred already in the young leaves, thus minimizing the mismatch between the syntheses of the two types of photosynthetic pigments. However, overwintering outdoor cultivation of durum and bread wheat did not lead to the expected recovery of a wild-type-like leaf chlorophyll content (Figure 1). More interestingly, the mutants underwent an effective long-term acclimation of the functional traits related to photosynthesis. In mature plants, the alleviation of the functional phenotype was evident from the restored ability to control the photosynthetic electron poise (Figure 2 and Figure 3), as well as the tendency to more regular spider plots of OJIP-derived parameters (Figure 5). Especially in durum-wheat mutants, but also in 32A, the importance of accumulating interchain electron carriers and PSI end acceptors, as represented by Sm, emerges even more strongly than in previous indoor experiments [16].

### 3.1. The Grain Yield per Plant Correlates with the Early Photosynthetic Parameters

The genetic lesion in chlorophyll-deficient wheat lines is attributed to a decreased activity of the Mg-chelatase, which impairs chlorophyll accumulation, particularly chlorophyll *b* [45,46,47]. Despite recurrent reports on highly productive chlorophyll-deficient crops, wheat included [9], the less-than-normal chlorophyll content in wheat mutants causes several alterations of the photosynthetic-membrane function and structure [14,15,16,20,48,49]. The picture we get from previous studies and the current research is that most of the efforts of the chlorophyll-deficient wheat plants are devoted to finding compensation for the genetic defect: this is certainly helped by the healthy homoeologous genes in another or other subgenome(s), but it also benefits from energy-consuming adjustments of the photosynthetic electron flow [15,16]. The consequence is that the chlorophyll-synthesis mutation has a pleiotropic effect, including significant long-term impacts on plant morphogenesis and reproduction. Accordingly, we show that in this collection of chlorophyll-depleted mutants, it is possible to discriminate between different degrees of productivity by performing fast and non-invasive chlorophyll-fluorescence measurements at the initial stages of plant development. Interestingly, input phenotyping data were mostly correlated with the synthetic parameter, “single-plant productivity” (Figure 7). This observation reveals that increasing degrees of chlorophyll depletion do not result in an unambiguously identifiable, single, impaired productive trait, such as the grains per ear or the ear length (see also [49]), but rather lead to an overall depression of the capacity to produce grains of the single plant. In the over-wintering experiment, a significant contribution to such decrease is due to the lowered tillering of the mutants, indicating the fact that, directly or indirectly, the genetic defect influences productivity by limiting the emergence of axillary shoots well before the reproductive phase (Figure 6). This response is interesting, because a low number of tillers is usually a shade-avoidance response in wheat cultivated in a dense canopy, and is counterbalanced by investment in plant height [50,51]. According to the principle equating a low-chlorophyll phenotype with an improved light interception [7,8], lower plants (a tendency visible in Table A3), but with many tillers, would have been expected. We can suggest that in chlorophyll-deficient wheat, the potential benefit of a higher light transmittance through the canopy is overcome by the energy-demanding adjustment of an altered photosynthetic electron transport chain.

### 3.2. PSII Is Less Photoprotected in the Mutants, but Moderate PSII Photoinhibition Can Help Control the Electron Flow into the Chain

Y(NO) and NPQ derive from dynamic-measuring protocols—the light-response curves—and are suited to a correlative analysis if expressed as synthetic phenotypic indexes. Given the different light-dependency of Y(NO) and NPQ in the wheat lines, we extracted Y(NO)_MAX_ and NPQ_MAX_ from the light curves (Figure 2 and Figure 3). The former proves a defective electron-transport control, the latter a reduction in thermal-dissipation capacity, i.e., two main physiological traits characterizing the mutants based on previous indoor experiments [14,15]. Y(NO)_MAX_ correlated negatively only with the number of grains per ear (Figure 7); therefore, in this outdoor experiment, Y(NO)_MAX_ did not appear to be a very straightforward index of the scope of the correlation. Y(NO) certainly changes with light intensity, but its maximum value, representative of a quasi-steady state, can be not very informative (see [15]). More interesting was NPQ_MAX_, which in young plants was clearly dependent on the phenotype severity, correlating with a single plant’s productivity (Figure 3). In the mutants, a lower capacity to develop NPQ is a physiological trait related to an insufficient proton pumping into the thylakoid lumen, which was interpreted as a side effect of the defective CEF [14,15]. It was previously shown that the parallel up-regulation of the ATP-synthase activity allows for a sufficient ability of these mutants to sustain ATP synthesis [15]. Despite the defective CEF, ensuring an adequate supply of ATP permits the stability of the photosystems, e.g., by supporting the repair of photodamaged PSII [25]. However, NPQ_MAX_, as a technical parameter is complex in origin, and includes not only pH-dependent de-excitation processes (“high-energy quenching”, qE), but also other components, e.g., related to the plastid redox state or the integrity of photosystems [38,52,53,54,55,56,57]. Not all NPQ components have a primary photoprotective meaning, and high levels of NPQ may not correspond to high levels of PSII photoprotection [58,59]. Interestingly, a comparison of NPQ_MAX_ and pNPQ makes it clear that the former may underestimate the severity of the photosynthetic-regulation defect, particularly in the “intermediate” mutants 7A and 7B, and also 4B (Figure 3). The phenotype alleviation occurring during the life cycle of outdoor-grown plants includes at least a partial recovery of NPQ_MAX_ to wild-type values, in one case—4B—even exceeding them (Figure 3A–D). However, a comparative analysis of the corresponding pNPQ invites us to be particularly cautious about the conclusions, because a reduced PSII-photoprotection capacity remains in 7B and 32A, which may be expected, but also in 4B, as a tendency (Figure 3). pNPQ was a robust phenotypic index, strongly correlating with the tillering degree and the single plant productivity (Figure 7).

Although a reduced PSII photoprotection is usually considered as negative for the plant, the results obtained with 1-qP invite us to rethink this concept in the chlorophyll-deficient wheat mutants. Unexpectedly, even the most severe young mutants presented a very good control of the fraction of closed PSII centres (Figure 3I–L). 1-qP describes the state of only the PSII centres that are reached by excitons and which can use them for the photochemical charge separation, therefore excluding the quenched and photoinhibited PSII centres [40]. The lesser degree of PSII photoprotection (lower pNPQ) associated with a strict control of 1-qP suggests that, in the mutants, the accumulation of photoinactivated PSII centres can help decrease the electron inflow into the chain. The regulatory relevance of moderate PSII photoinhibition has been proposed in other systems in which PSI can be particularly prone to photodamage [60,61], which is also the case of the chlorophyll-deficient wheat mutants [14,20].

### 3.3. The Accumulation of Interchain Electron Carriers Can Help Compensate for the Membrane Over-Reduction in Mutants, Particularly of Durum Wheat

Single-plant productivity in wheat was correlated with three JIP-test-derived parameters, catching different facets of the light-energy conversion in the mutants: ABS/RC, TRo/RC and Sm (Figure 7). ABS/RC and TRo/RC are energy-flux parameters describing the functional PSII antenna size and the maximum trapped energy in PSII [41,42]. The downsizing of the PSII antenna was an expected consequence of the reduced availability of chlorophyll *a* and *b* for assembling the LHCII, shared by all mutants at their young stage (Figure 5A,B). The reduced trapping ability in active PSII was also expected, and consistent with the smaller antenna of the photosystem, despite a tendency of the mutants to higher φPo, which is, however, due to a lower PSI/PSII ratio rather than a genuine increase in PSII photochemical activity [16,20]. In the previous indoor experiment, ABS/RC tended to very effectively recover wild-type values, making it a weak phenotyping index [16]. In the case of the outdoor, over-wintering wheat growth, the reduction of the PSII antenna size appears to be a more persistent trait, perfectly in line with the severity gradient in the chlorophyll *a*/*b* ratio still observed in mature plants (Figure 1). It was previously proposed that the chlorophyll-deficient wheat lines may have an intrinsic ability to adjust the PSII antenna system under the unfavourable supply of chlorophylls [16]. This regulation, whose molecular backgrounds are unknown, may help mitigate an unbalanced excitation distribution between PSII and PSI. Evidently, outdoor cultivation is less permissive to such regulation, which occurred effectively only in 4A and 4B (Figure 5). In other mutants, we observed instead an enhanced accumulation of electron transporters. A similar response was observed in the previous indoor experiment, but it was more linked to the relative pool size of the end acceptors of PSI (ferredoxin and FNR), represented by δREo, than to the overall abundance of electron carriers, Sm [16]. Indoors, Sm values were indicators of the chlorophyll-depleted phenotype severity in bread-wheat mutants, but not in the durum lines [16]. Outdoors, the increase in Sm occurred in all young mutant plants (Figure 5A,C). Subsequently, while Sm tended to revert to wild type in bread-wheat mutants, the differences between durum-wheat genotypes were magnified at the end of the season, up to the excessive Sm increase characterizing 7B (Figure 5B,D). A compensative reason, against the membrane over-reduction, was assigned to the seeming excess of end transporters [16], supported by findings about the role of FNR as a buffer for electrons to keep PSI in a safe oxidized state and prevent the accumulation of reactive oxygen radicals [62,63]. A relevant hypothesis, denoted as the “ultrastructural control” of photosynthetic electron transport, was proposed by Gu and co-workers (2022) [64]. They attribute the role of electron reservoirs to all mobile electron carriers, i.e., plastoquinone, plastocyanin and ferredoxin, to cushion the light fluctuations causing variations in the thylakoid-lumen swelling [64]. The electron-buffering activity demonstrated for FNR [63] suggests the inclusion of this enzyme in the model. In the JIP test, the concept of Sm covers all mobile pools and the enhancement of Sm in the mutant wheat lines can be interpreted consistently within the ultrastructural-control hypothesis, to offer an extended buffering system against the recurrent electron bursts caused by light flecks in a fluctuating-light regime [16]. In a natural environment, where the light fluctuations are combined with other variables and the re-sizing of the PSII antennae cannot be reached completely, the Sm increase emerges as possibly the main compensation mechanism of the electron-flow defects in chlorophyll-deficient mutants. More intense exploitation of this response in durum wheat than in bread wheat is in line with the acclimative advantage offered by hexaploidy in the latter [14,65]. In most mature plants, the achievement of an electron-flow regulation similar to the wild-type lines, as seen for Y(NO), testifies to the success of the defect compensation. However, the investment in electron carriers is energy-demanding, and conceivably diverts a part of the metabolic energy from vegetative growth and reproduction to the acclimation of photosynthesis. Therefore, a strong negative correlation links the single-plant productivity and Sm, which can be proposed as a robust potential marker in plant-phenotyping projects in the open field, and aims to increase grain production in chlorophyll-deficient durum- and bread-wheat lines. For example, the early screening of chlorophyll-deficient mutants for excessive Sm may help discard low-productive lines.

Although not correlating with plant productivity, the JIP-test parameter DIo/RC was the most affected in young plants. The lack of correlation is due to the absence of a clear correspondence with the phenotype-severity series. Two factors concur in lowering the DIo/RC—the reduced PSII antenna size and the seemingly increased PSII photochemical activity—making it difficult to assess whether the DIo/RC decrease can be truly symptomatic of a reduced capability for energy dissipation in PSII unit [66].

## 4. Conclusions

In the semi-field overwintering experiment, the chlorophyll-deficient wheat mutants tend not to recover a wild-type-like phenotype with respect to the leaf chlorophyll content and PSII antenna size (ABS/RC). In contrast, the photosynthetic-membrane functionality, which is affected by the deregulated electron flow, tends to revert to the wild type, aided by mechanisms that counteract the membrane over-reduction. The tillering phase had a substantial effect in reducing the single plant’s grain productivity, indicating that the mutation itself, or the energy-consuming compensatory mechanisms, significantly influence plant morphogenesis. The correlation analysis between early photosynthetic phenotype and grain yield per plant suggests that some chlorophyll fluorescence-derived parameters are good candidates for future phenotyping applications in the open field, particularly pNPQ, ABS/RC and Sm.

## 5. Materials and Methods

### 5.1. Plant Material

The wheat lines used in this work were the same as described by Živčak et al. [14]. Four lines were used for *Triticum durum* L.: the wild-type LD222 and three derived chlorophyll-depleted mutants, namely ANDW-7A, ANDW-8A and ANDW-7B. Similarly, four lines were used for *T. aestivum* L., the wild-type Novosibirskaya 67 (NS67) and three derived chlorophyll-depleted mutants, namely ANBW-4A, ANBW-4B and ANK-32A. The main characteristics of the wheat lines are reported in Table A1.

### 5.2. Study Site and Experiment Design

Plants were sown in parcels (30 seeds per parcel) at the Botanical Garden of the University of Ferrara (44.841912 N, 11.622454 E) in October, for two subsequent seasons (2020–21 and 2021–22), following the regular schedule for sowing *T. durum* and *T. aestivum*, and therefore allowing the plants to overwinter and go through the tillering phase in outdoor conditions. The general meteorological conditions, including solar radiation, are publicly available through the website of the Meteorological Observatory of the Botanical Garden of Ferrara (http://www.meteosystem.com/dati/ortofe/index.php, accessed on 25 November 2022), and a graphical summary of the weather features during the experimental seasons is reported in Appendix D (Figure A1). The parcels were exposed to direct sunlight for a major part of the day, but also subjected to significant light fluctuations (Figure A2). The first round of analyses was performed during the first weeks after plant germination, following the appearance of the third leaf in all lines. The second round of analyses was performed on the flag leaf during the second week of May.

### 5.3. Morphometric Measurements and Grain-Productivity Estimation

Morphometric indexes consisting of plant height, spikelet number per ear, ear length, and tillering degree were measured after the complete emergence of the ear. At the end of the season, mature ears were harvested, and grain productivity was estimated by counting the number of grains in each ear with ten random replicates per genotype. The grains of each ear were counted and weighed, to obtain the average number of grains per ear and the average weight of a single grain. The synthetic parameter and best estimator of plant productivity, namely “grain productivity per plant”, was calculated by multiplying the average weight of a single grain by the number of grains per ear and the tillering degree.

### 5.4. Quantification of Photosynthetic Pigments

Leaf segments (2 cm^2^) were cut from the 3rd leaf of young plants and the flag leaf of mature plants. The segments were subsequently weighed, reduced to smaller pieces, and placed in glass tubes containing 3 mL of 80% (v/v) acetone buffered with HEPES-KOH (pH 7.8). Extraction was performed at −20 °C for 3 days, until complete depigmentation of the leaf samples. Extracts were analysed using a spectrophotometer, Ultrospec 2000 (Pharmacia Biotech, Piscataway, NJ, USA); chlorophyll-*a* and *b* content in the extract were determined, according to Ritchie [67], while total carotenoids were calculated using the equation by Wellburn [68].

### 5.5. Modulated-Chlorophyll-a-Fluorescence Measurements

Modulated-chlorophyll-*a* fluorescence was measured in leaves that were cut and immediately transported to the laboratory. The leaves were then dark-acclimated for 30 min, and analysed using a Junior-PAM (Heinz Walz, Effeltrich, Germany) fluorometer. Measurements were performed as described by Colpo et al. [59], with some modifications. Minimum and maximum fluorescence (*F_0_* and *F_M_*, respectively), were measured before starting the light-response-curve experiment. Ten irradiance intervals (actinic light from 65 to 1500 µmol photons m^−2^ s^−1^), each with 7 min length, were applied to the sample. To avoid leaf dehydration, the samples were placed on top of a damp piece of filter paper during the whole measuring routine. Saturation pulses were 0.6 s long, to determine *F_M_’*, each followed by 5 s exposure to far-red light, to determine *F*_0_*’.* Quantum yields of actual PSII photochemistry [Y(II)], non-regulatory energy loss [Y(NO)], and regulatory thermal dissipation [Y(NPQ)] were calculated, according to Hendrickson et al. [30]. NPQ was calculated as (*F_M_−F_M_’*)/*F_M_’*, and qP as (*F_M_’−F*)/(*F_M_’−F_0_’*), where *F* is the fluorescence level recorded just before applying the saturation pulse [32]. Photoprotective thermal dissipation (pNPQ) was calculated by applying the qP_d_ protocol, as described by Ruban and Murchie [39], with modifications as detailed by Colpo et al. [59]. The theoretical fundament of the “qP_d_ model” is based on the comparison between actual *F’*_0_ and the theoretical *F’*_0_ value, calculated according to Oxborough and Baker (*F’*_0 *calc*_) [69]. Since the latter is insensitive to PSII photoinhibition, their comparison at the different steps of a light-response curve allows one to verify the presence of PSII photoinhibition and its extent [39]. The *F’*_0_-derived qP_d_ parameter lies between 0 (all PSII are photoinactivated) and 1 (all PSII are active). In the model, the threshold used to assess the occurrence of PSII photoinhibition was set at qP_d_ = 0.98; the NPQ associated with the last qP_d_ > 0.98 is pNPQ [39]. In the case where qP_d_ assumes values above 1 because of significant PSII-antenna uncoupling, the determination of pNPQ is still possible, with suitable corrections [59,70].

### 5.6. Fast-Chlorophyll-a-Fluorescence Measurements

Fast-chlorophyll-*a* fluorescence was measured directly in the field on the 3^rd^ or flag leaf, according to the growth stage. After 30 min of dark acclimation in the leaf clip, the fluorescence emission was recorded using a portable HandyPEA (Hansatech Instruments, King's Lynn, Norfolk, UK) fluorometer. For analysis, a 1-s-long saturating pulse was applied to the sample with an intensity of 3500 μmol photons m^−2^ s^−1^ (650 nm light provided by the LEDs integrated in the measuring head of the device). The derived OJIP transients were analysed according to the energy fluxes model as reported by Strasser et al. [41], Stirbet and Govindjee [42], and Tsimilli-Michael [71], allowing the calculation of selected basic and derived parameters with the Biolyzer software (Fluoromatic Software, Geneva, Switzerland). *F*_0_ value was sampled at 20 µs. The list of the parameters and their phenomenological meaning is reported in Appendix B, Table A2.

### 5.7. Data Analysis and Correlation Matrix

Data obtained from each set of measurements were analysed using the Microsoft Office Excel^TM^ 365 (Microsoft) software for the preliminary elaborations. Graphs and statistical analyses were carried out using Origin™ version 2022 (OriginLab, Northampton, MA, USA). Statistical comparison between the replicates of each group (durum and bread wheat) was performed using ANOVA one-factor test with α = 0.05 significant threshold, followed by the post hoc Tukey’s test (α = 0.05) for means comparison. The Pearson’s *r* correlation matrix between productivity values, morphometric indexes, and photosynthetic phenotyping data was built with Microsoft Office Excel.

## Figures and Tables

**Figure 1 plants-12-00822-f001:**
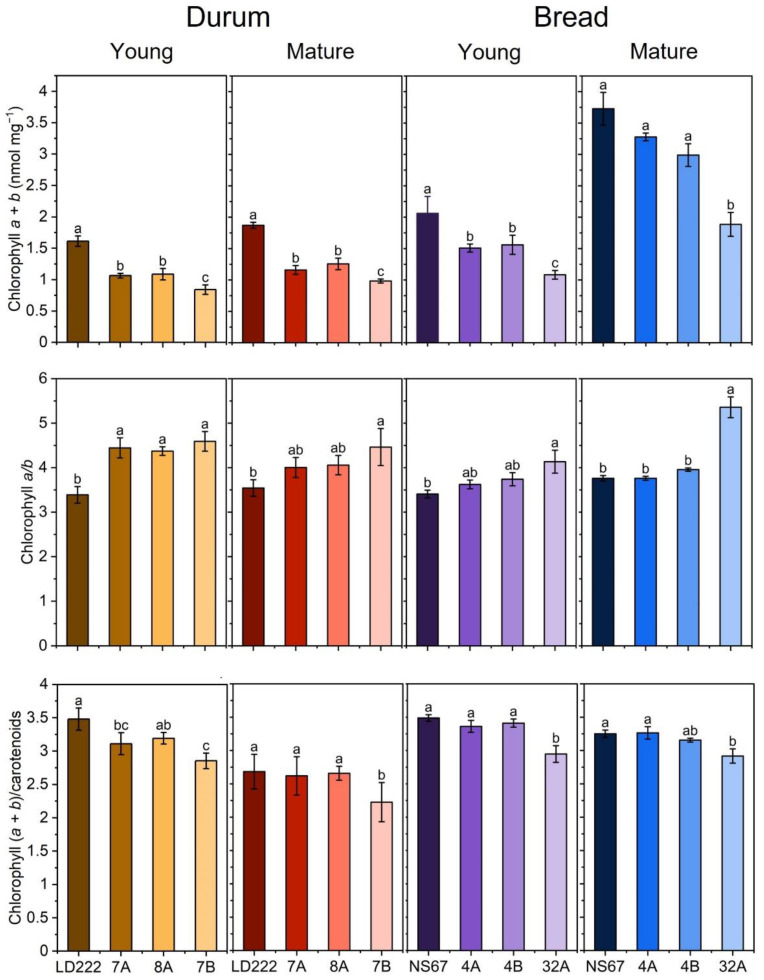
Photosynthetic-pigment quantification in leaves of young and mature plants of durum and bread wheat. Histograms represent average values ± Standard Deviation for n = 5–8; different letters indicate a significant difference at *p* < 0.05, as determined using one-factor ANOVA followed by Tukey’s test.

**Figure 2 plants-12-00822-f002:**
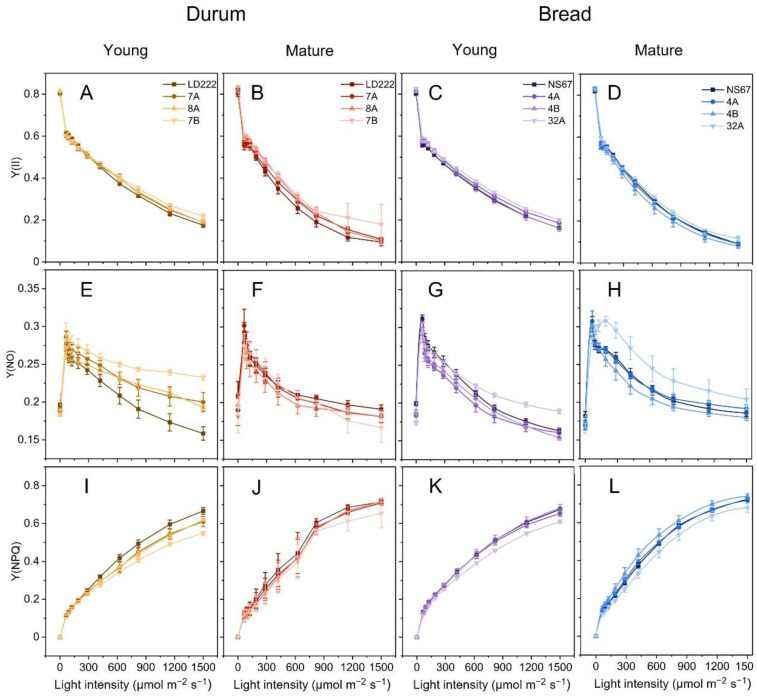
Long-term light-acclimation features of photosynthesis in young and mature bread-wheat plants. Light-response curves relative to: (**A**–**D**) actual quantum yield of PSII photochemistry Y(II), (**E**–**H**) quantum yield of constitutive, non-regulatory energy dissipation Y(NO), and (**I**–**L**) quantum yield of regulatory thermal dissipation Y(NPQ). The curves were recorded during 60 min of exposure to increasing actinic-light intensities. Average values ± Standard Deviation for n = 4–6.

**Figure 3 plants-12-00822-f003:**
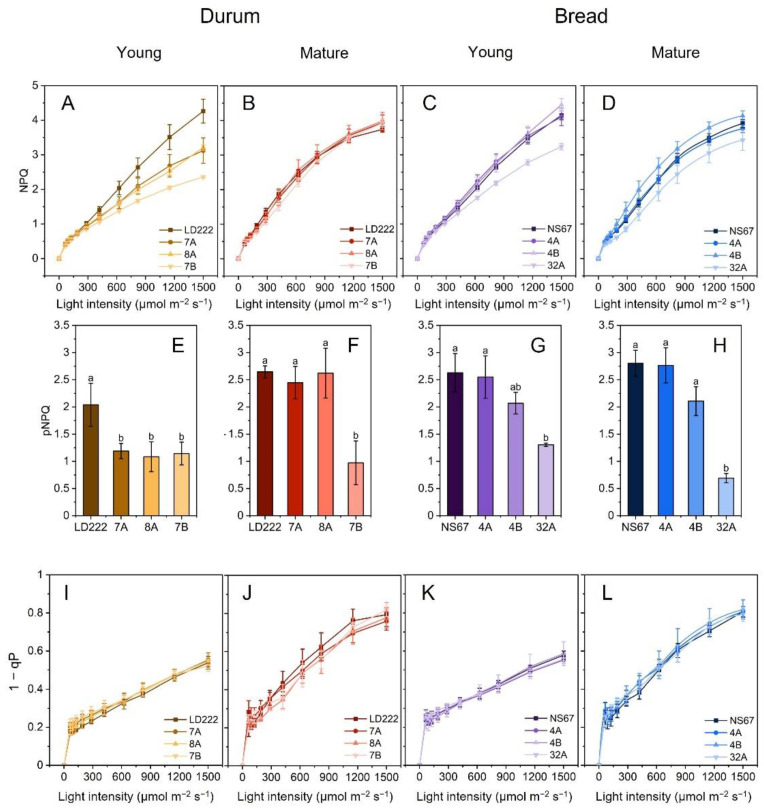
Fluorescence-quenching analysis in young and mature plants of bread and durum wheat. (**A**–**D**) NPQ-light-response curves of young durum (**A**), mature durum (**B**), young bread (**C**) and mature bread (**D**). (**E**–**H**) Photoprotective NPQ (pNPQ) values obtained after applying the “qP_d_ method”. Histograms represent average values ± Standard Error for n = 4–6; different letters indicate a significant difference at *p* < 0.05, as determined using one-factor ANOVA followed by Tukey’s test. (**I**–**L**) Light-response curves of 1-qP of young-durum (**I**), mature-durum (**J**), young-bread (**K**) and mature-bread (**L**) plants. All light curves were obtained during 60 min exposure to increasing actinic-light intensities. Average values ± Standard Deviation for n = 4–6.

**Figure 4 plants-12-00822-f004:**
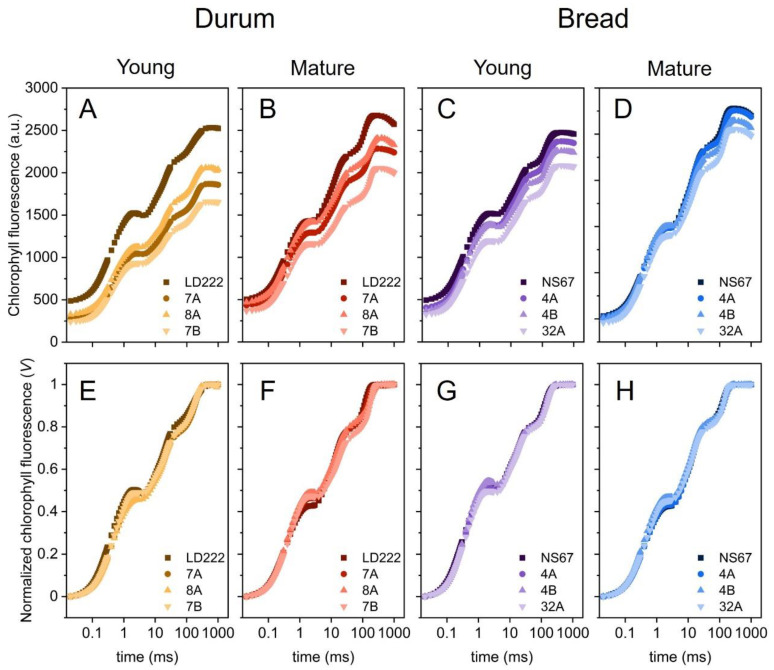
Fast-chlorophyll-*a*-fluorescence emission in young and mature plants of durum and bread wheat. (**A**–**D**) Average OJIP transients on logarithmic timescale. (**E**–**H**) Average OJIP-transient normalized between *F*_0_ = *F*_20*µs*_ and *F_M_* on logarithmic timescale.

**Figure 5 plants-12-00822-f005:**
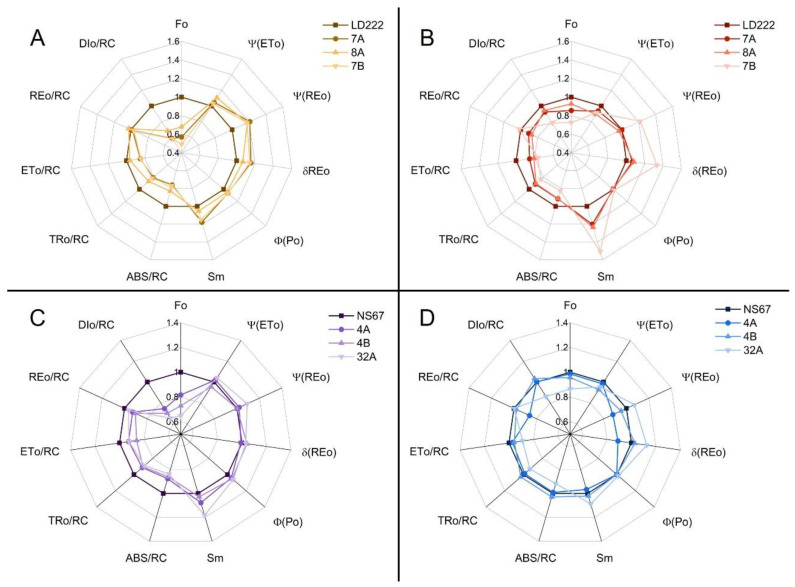
Spider plots of selected fast-chlorophyll-*a*-fluorescence-derived parameters calculated according to the “Energy Fluxes” model in young and mature leaves of durum- and bread-wheat plants. Each point represents the average value normalized on the respective average value of the wild-type line (LD222 in durum wheat, NS67 in bread wheat) for n = 8–12 replicates. (**A**,**B**) Spider plots of young (**A**) and mature (**B**) durum-wheat lines. (**C**,**D**) Spider plots of young (**C**) and mature (**D**) bread-wheat lines. For the definition of the parameters, see Appendix B.

**Figure 6 plants-12-00822-f006:**
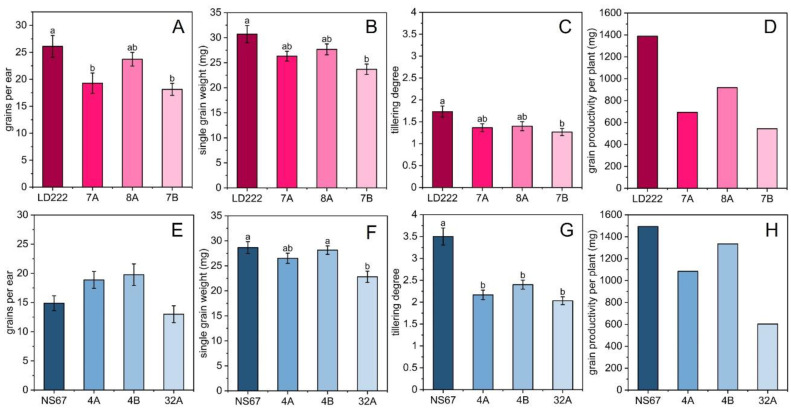
Productivity parameters measured at the end of the season in durum- and bread-wheat mature plants. (**A**,**E**) Average number of grains per ear (n_ears_ = 10). (**B**,**F**) Average weight of a single grain harvested from the 10 ears used in the previous measurement. (**C**,**G**) Tillering degree is the average tillering degree in durum and bread wheat (n = 30). Histograms represent average values ± Standard Error; different letters indicate a significant difference at *p* < 0.05, as determined using one-factor ANOVA followed by Tukey’s test. (**D**,**H**) Grain productivity per plant, a synthetic parameter that combines the previous three and estimates the mass of grains that was produced by a single plant for each wheat line.

**Figure 7 plants-12-00822-f007:**
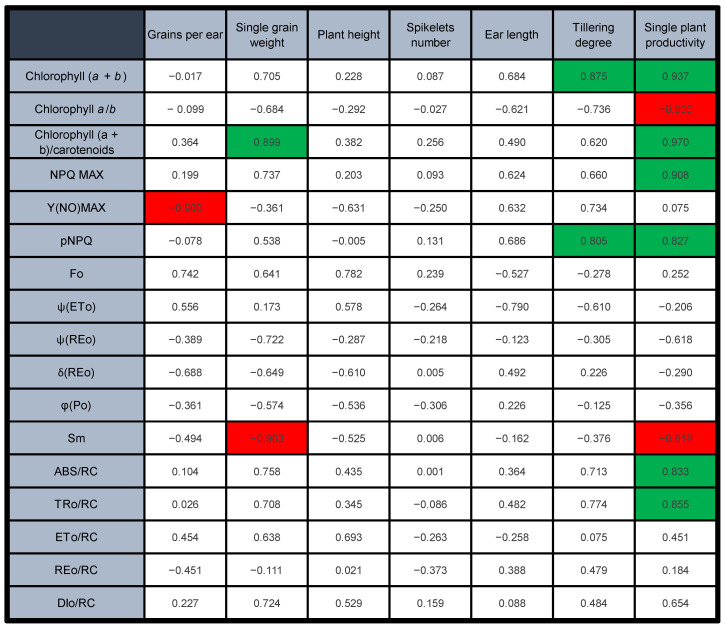
Matrix of Pearson’s *r* correlation coefficients of photosynthetic parameters vs. productivity parameters in durum- and bread-wheat lines. Correlations were constructed with average values of the measurements performed in the young plants using Microsoft Office Excel^TM^ (Microsoft). Green colour marks significant positive correlations, red the negative (*p* < 0.05).

## Data Availability

Raw data that support the findings of this study are made available on request.

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
