# Peer review of "Long-Term Alleviation of the Functional Phenotype in Chlorophyll-Deficient Wheat and Impact on Productivity: A Semi-Field Phenotyping Experiment"

_plants, 2023, doi:10.3390/plants12040822_

Round 1

Reviewer 1 Report

In the present work the photosynthetic acclimation of wheat mutants with reduced chlorophyll in a semi-field experiment was investigated. Additionally, the research considered the possible evaluation of phenotype traits as predictive parameters for plant productivity. Such an approach has extensively been studied by the same research team through indoor experiments, with very interesting results, published previously. Consequently, the novelty of the present work consists in the hypothesis that acclimation procedures may be more or less efficient under naturall conditions mainly by means of the significantly longer maturation period.

The initial hypotheses are clearly stated and either confirmed or not, the authors provide satisfactory and extended comments based on bibliography and their previous significant experience on this subject. The design of experimentation and the methodology is the appropriate and items are thoroughly discussed. Consequently, I would suggest acceptance after a minor revision regarding only one comment on the results section as follows:

lines: 205-207 (Results). Authors claim: A tendency to slightly  higher Y(II) was visible especially in the most severe mutants and in the young leaves, in  line with previous reports [9-11] (Figure 3A,C. In mature plants, this effect on Y(II) was  still observed in durum wheat, while in bread wheat, the curves were nearly overlapping  (Figure 3B,D)). However, to my opinion, the almost identical light curves do not support this claim. Although, this particular result is not critical for the rest of the discussion, I think that should be corrected.

Reviewer 3 Report

Authors need to improve goal definition in the introduction and improve clarity of some terms and expression throughout the manuscript. It is sometimes difficult to make some judgements on some information, if they are expressed with no-scientific term and in adjective. This is appearing throughout the manuscript.

Somehow I am missing Materials and Methods parts. There is 1 page of methods at the end, nothing surprising. Maybe I am missing some information about data manipulation in ths section. The article goes directly from Introduction to Results, what is unusual. Therefore I am not able to judge, how were the samples manipulated and distinguish between what is related to indoor and outdoor sample manipulations and benefits from this. It was proclaimed in Intoduction.

Results description is too long. Probably only key features of the results should be presented. I mostly run across the figures.

Line 43 – I do not understand to what should the intermediate step serve. The structure of sentence is not good. We either study some processes in closed environment, in which we are able to control most of things, or measure things in the wild, with limited control of environment, what affects the result. What would be the meaning of crossing then, if we are interested in some physiology related processes. By the first look, we can either measure plants grown in controlled environment in wild, or compare control and wild plants, maybe that should be somehow highlighted at the start.

Line 47 – characterization of photosynthesis is not very good expression for something we are doing. I would prefer more specific expression, with what is meant by characterization. That would require new expression and set of sentence, or remove the sentence.

Line 70 – I would prefer information, how lowering of chlorophyll content in upper leaves can be or is achieved. Are these those mutants, then the information should be prioritized, because introducing structural mutants only introduce confusion.

Line 130 – I am not very familiar with terms alleviation and exacerbate, the words look too uncertain to me, does it mean good news or bad news in context of research, I am really not sure. May be easier expression would be better. The expression, which would express the change in relation to chlorophylls or yields.

Line 140 – last paragraph of introduction should be definition of research goal. Here I am detecting description of some results, which however is not somehow cited, or there is no indication, how to work with the information. From this point definition of goals is not very convincing, and I was not able to discriminate hypothesis and goal of the article at all. It should be clear statement. The overall confusion is related to those alleviation and exacerbate terms, which are difficult to understand.

Line 441 - recover a wild-type phenotype with respect to the leaf chlorophyll content, emphasizing the impact of the genomic context – does it mean that mutants grown inside in controlled environment have similar chlorophyll to control non-mutant plants? But what I understand the article does the opposite – growing mutants outside and seeing recovery, and effects on productivity.

Line445 – I do not understand sentence with susceptibility and growth relevance, these two terms are reasons why I do not understand it. The terms are again too unspecific and uncertain.

Line 465 – similar term “predictive power of the early biochemical and functional traits” is not very conclusive for discussion, reader is not sure to what “predictive power” is related. Whether to final grain yield or to photosynthesis. I recommend to delete these sentences or be more specific in terms to which the information is related. But the rest of the paragraph looks fine; this is usually appearing at the start of paragraphs, where often not very specific information is given.

Line 552 - Prompt-chlorophyll a fluorescence parameters – Prompt is too informal word for science literature.

Line 609 – relief in mature plants refers to what? Sounds to me like some pain recovery, it could be also rewritten.

The conclusion seem to me too generic, information in conclusion should be more specific, showing major achievements of the study. Similarly I am again not able to fully understand content of the conclusion because of unspecific terms like mitigation, marginally, compensatory mechanism two times, defective control, less efficient acclimative potential.

Round 2

Reviewer 3 Report

I have checked the critical parts in the manuscripts, and it looks ok. Results description seems to e modest now, so I agree with releasing the article.